# Zika M—A Potential Viroporin: Mutational Study and Drug Repurposing

**DOI:** 10.3390/biomedicines10030641

**Published:** 2022-03-10

**Authors:** Prabhat Pratap Singh Tomar, Miriam Krugliak, Anamika Singh, Isaiah T. Arkin

**Affiliations:** Department of Biological Chemistry, The Alexander Silberman Institute of Life Sciences, The Hebrew University of Jerusalem, Edmond J. Safra Campus Givat-Ram, Jerusalem 91904, Israel; ppstdbt@gmail.com (P.P.S.T.); miriamkru@savion.huji.ac.il (M.K.); anamika.iitr7@gmail.com (A.S.)

**Keywords:** flavivirus, Zika virus, West Nile virus, Dengue virus, membrane glycoprotein, ion channel proteins, viroporins, drug discovery, drug repurposing

## Abstract

Genus Flavivirus contains several important human pathogens. Among these, the Zika virus is an emerging etiological agent that merits concern. One of its structural proteins, prM, plays an essential role in viral maturation and assembly, making it an attractive drug and vaccine development target. Herein, we have characterized ZikV-M as a potential viroporin candidate using three different bacteria-based assays. These assays were subsequently employed to screen a library of repurposed drugs from which ten compounds were identified as ZikV-M blockers. Mutational analyses of conserved amino acids in the transmembrane domain of other flaviviruses, including West Nile and Dengue virus, were performed to study their role in ion channel activity. In conclusion, our data show that ZikV-M is a potential ion channel that can be used as a drug target for high throughput screening and drug repurposing.

## 1. Introduction

Flaviviruses are a positive-sense, single-stranded RNA virus family that contains several important human pathogenic viruses. It includes dengue virus (DenV), Japanese encephalitis virus (JEV), tick-borne encephalitis virus (TBEV), West Nile virus (WNV), yellow fever virus (YFV), and Zika virus (ZikV) [1,2,3]. Such viruses pose a serious threat to human health and have the potential to cause future epidemics and pandemics. Reports that have surfaced recently about potential outbreaks include DenV, the recent ZikV outbreak in South America, YFV outbreaks in Africa and Brazil, and the spread of WNV across North America [4,5,6,7].

Zika Virus is an enveloped flavivirus with icosahedral symmetry. The virus’s genome comprises ca. 11 kb long positive-sense single-stranded RNA. It encodes three structural proteins, namely: capsid (C), precursor membrane (prM), envelope (E), and seven non-structural proteins (NS) [8]. The virus transmission to humans is mainly by the Aedes mosquito, but there is a possibility of human-to-human transfer [9,10,11]. ZikV causes congenital neurological malformations, known as Congenital Zika Syndrome (CZS), and is responsible for disabilities in children born to infected mothers during pregnancy [8,9,10,11,12,13,14,15,16,17]. Consequently, it has been declared a “Public Health Emergency of International Concern” by the WHO [18].

Precursor membrane, prM, one of the most extensively studied structural proteins from flaviviruses, plays an important role in the viral infectivity cycle and its release. After synthesis, prM protein protects premature fusion of the E protein with transport vesicles at low pH [10,19]. Before the virus’s release, a furin-like protease from the host cleaves prM into matured M protein in the trans-Golgi network [20,21]. The cleavage of prM into M is required for membrane fusion between the viral and host endosomal membranes as it triggers rearrangement of the E protein and maturation of the virus [19].

Phylogenetic analyses of ZikV identified African and Asian lineages, with the latter responsible for the epidemics detected so far [13]. However, the African strain MR766 is known for its higher virulence. It is reported that when the prM gene of MR766 is replaced with the Asian strain PRVABC59, its lethality is reduced in IFNα/β receptor knockout (IFNAR${}^{-/-}$) mice [22]. This finding is associated with reduced neuro-invasiveness of the virus after subcutaneous infection [22]. The higher neuro-invasiveness of the African variant has been linked to changes in highly conserved amino acids that affect the positive charge and hydrophobicity of the exposed surface of the protein. These conserved amino acids are related to the higher capability of the virus to cross the blood-brain barrier [22].

The search for vaccines and drugs against ZikV is still in progress. Few vaccines and medications are in phase I and phase II clinical trials [15,22,23]. There are also vaccine candidates in non-clinical stages based on prM-E antigen [21]. Still, there is no approved vaccine or medicine for Zika available for use [24,25,26].

Viroporins are a family of small, hydrophobic integral membrane proteins that possess channel activity [27]. The M2 channel from the Influenza A virus is one of the most well-characterized viroporins. Inhibition of the M2 channel by amantadine and rimantadine curb the viral infectivity [28], thereby demonstrating the potential of viroporins to serve as drug targets. Similarly, the functions of flavivirus M proteins as an ion channel were reported in earlier studies of DenV and WNV [29,30].

Herein, we focussed on the characterization of matured Zika virus M protein as a potential viroporin utilizing three different bacteria-based assays. The mutational study of the transmembrane region of matured membrane glycoprotein M protein sheds light on the effect of conserved amino acids residues among ZikV, WNV, and DenV. Finally, drug repurposing screening efforts have yielded several blockers, demonstrating that ZikV-M could be an attractive target to curb the infectivity of the virus.

## 2. Materials and Methods

### 2.1. Sequence, Cloning, and Bacterial Strains

Gene sequences for matured ZikV-M (YP_009227197.1), Dengue virus membrane glycoprotein (NP_722459.2), and West Nile MgM (NP_776013.1) were retrieved from NCBI. G-blocks, containing the following genes with a linker, histidine tag, and a stop codon at the C-terminal were ordered from Integrated DNA Technologies (IDT; Coralville, IA, USA). Genes were cloned into the pMAL-p2X plasmid (New England Biolabs; Ipswich, MA, USA), using Gibson Assembly (New England BioLabs; Ipswich, MA, USA) in continuation of the open reading frame of the maltose-binding protein (MBP). The plasmid was subsequently transformed into ultra-competent cells [31], and the positive clones were confirmed by sequencing. Multiple sequence alignment of the proteins was done using Clustal Omega [32]. The visualization of the alignment was conducted by ESPript 3.0. [33]. The cryo-EM structures (PDB id- 6CO8, 5IRE [34,35]) of ZikV-M were taken as a template for generating the model of the transmembrane domain of ZikV-M.

Four bacterial strains of *Escherichia coli* were used in the study: (i) DH10b (purchased from Invitrogen; Carlsbad, CA, USA); (ii) LB650 (ΔtrkG, ΔtrkH, and ΔkdpABC5 [36] were a kind gift from K. Jung, Ludwig-Maximilians Universität München and G.A. Berkowitz, University of Connecticut; NT326 (ΔmalE) were a kind gift of D.M. Engelman of Yale University; and LR1 (chromosomal pH-sensitive GFP [37], a kind gift from M. Willemoës and K. Lindorff-Larsen, University of Copenhagen [38]. As mentioned in previous studies, the different strains were used for the negative genetic assay, positive genetic assay, maltose complementation assay, and proton conductivity assays [30,39,40,41,42,43,44,45].

### 2.2. Western Blot

For three hours, the ZikV-M MBP chimera was expressed in DH10B cells at 37 ${}^{\circ }$C after induction with 0.1 mM β-d-1-thiogalactopyranoside (IPTG). Uninduced cells were used as control. Cells were calibrated for O.D._600 nm_ = 1. The over-expressed cells were pelleted down at 3500 rpm for 10 min at 4 ${}^{\circ }$C. The cell pellet was lysed using lysis buffer (40 mM NaCl; 2.7 mM KCl; 10 mM Na${}_{2}$HPO${}_{4}$; 1.8 mM KH${}_{2}$PO${}_{4}$; pH 7.4) containing 100 mM PMSF, 100 µg/mL Dnase and 0.2 mg/mL lysozyme. The cells were kept at 37 ${}^{\circ }$C for 15 min on a dry bath and then freeze-thawed twice immediately followed by ultrasonication for five cycles of 10 s with a 10-s gap at 40 watts (Vibra-Cell™ Ultrasonic Liquid Processors; Newtown, CT, USA). Cleared cell lysates were collected after centrifugation (12,000 rpm for 20 min, 4 ${}^{\circ }$C) and further resolved using electrophoresis followed by transfer of the antigens to nitrocellulose membranes using a Semi-dry Trans-Blot^®^ Turbo™ Transfer System (BIO-RAD; Hercules, CA, USA). The membranes were blocked with TBS-T (tris-buffered saline containing Tween-20) containing 1% low-fat milk, incubated overnight with a primary antibody (Anti-His, New England BioLabs; Ipswich, MA, USA), washed three times with TBS-T, incubated for 60 min with a secondary antibody linked to horseradish peroxidase (anti-rabbit, Novagen; Temecula, CA, USA) and washed once again with TBS-T. Immunoreactive bands were detected using an ECL kit (Biological Industries; Haemek, Israel).

### 2.3. Maltose Complementation Assay

A plasmid containing the ZikV-M chimera was transformed into NT326 cells (ΔmalE) [46]. An empty pUC-19 plasmid with ampicillin resistance was used as a control. Transformed cells were grown on M9 agar plates containing 0.4% maltose as the sole carbon source with 100 µM IPTG for 72 h at 37 ${}^{\circ }$C [47,48].

### 2.4. Negative Genetic Assay

DH10B cells containing the ZikV-M plasmid were grown overnight in LB media (100 µg/mL ampicillin and 1% D-Glucose). The secondary culture was obtained by diluting the starter inoculum (1:100) in the aforementioned media. The culture was grown up to O.D._600 nm_ = 0.2 and then used for the assay with IPTG concentrations ranging from 12.5–100 µM in 96-well-plates. The plates were placed in an Infinite M200 pro reader (Tecan; Männedorf, Switzerland) at 37 ${}^{\circ }$C overnight with constant shaking at minimum speed. O.D._600 nm_ readings were recorded every 20 min. The same protocol was used for ZikV-M, WNV MgM, and DenV MgM mutants in DH10B. The pMal p2X plasmid (New England BioLabs, Ipswich, MA, USA) expressing only MBP was used as a control. All experiments were performed in duplicates or triplicates.

### 2.5. Positive Genetic Assay

Starter and secondary cultures of LB650 bacteria with ZikV-M plasmid were obtained as discussed above for DH10, but LB media is replaced by LBK media (100 mM KCl instead of NaCl). The secondary culture pellet was washed three times with LBK media and then dissolved in LBK media. The final culture was then used for the assay with different IPTG concentrations, with and without 5 mM KCl. Every 20 min at O.D._600 nm_, readings were recorded using Infinite M200 pro reader (Tecan, Männedorf, Switzerland) at 37 ${}^{\circ }$C overnight. All experiments were performed in duplicates or triplicates.

### 2.6. Proton Conductivity Test

LR1 cells were grown in the same conditions as DH10B cells. After 3 h of secondary culture, protein expression was induced by adding 100 µM IPTG at 37 ${}^{\circ }$C for 2 h. The cells were then washed twice and dissolved in buffer (200 mM Na${}_{2}$HPO${}_{4}$, 0.9% NaCl adjusted to pH 7.6 with 0.1 M citric acid, 0.9% NaCl) to an optical density of 0.25 at 600 nm. 200 µL of dissolved cells were put in 96-well optical black plates (Thermo Scientific, Waltham, MA, USA) with 30 µL buffer (or buffer + Drug/DMSO). 230 µL buffer was used for baseline reduction. Uninduced bacterial culture was used as a control. Fluorescent measurements were carried out after adding 70 µL of citric acid (300 mM, 0.9% NaCl) by a liquid handling system (Tecan; Männedorf, Switzerland). The plate reader was set for 390 nm and 466 nm excitation filters paired with a 520 nm emission filter. The wells’ emissions were read out, alternating between two filter pairs for 60 s. The proton flow was calculated according to the procedure described previously [38,49]. All experiments were performed in duplicates or triplicates.

### 2.7. Structure Modeling and Mutagenesis

The modeling of the ZK M has been performed using Phyre 2 server [50] and TMHMM 2.0 online tool [51,52,53] based on the Cryo-EM structure (PDB ID-5IRE, 6CO8 [34,35]). Primers for mutagenesis were designed using SnapGene software (Insightful Science; San Diego, CA, USA). Mutations were done with the quick-change lightning site-directed mutagenesis kit (Agilent; Santa Clara, CA, USA). Mutated Plasmids were transformed and confirmed by sequencing.

### 2.8. Drug Screening

Membrane Transporter/Ion Channel Compound Library (HY-L011, MedChem express; Monmouth Junction, NJ, USA) with 372 compounds and Drug Repurposing Compound Library (HY-L035, MedChem Express; Monmouth Junction, NJ, USA) with 2839 compounds were used for the screening. According to the vendor (MedChem Express), “the library contains approved drugs and compounds that have passed phase I clinical drugs, which have been completed extensive preclinical and clinical studies and have well-characterized bioactivities, safety and bioavailability properties”. The average compound molecular weight is 430 Da with a standard deviation of 277 Da. The reader is referred to MedChem Express for a complete list of the chemicals and their detailed description [54].

The high throughput screening method was used for the initial screening of compounds by a Tecan robotic system (Männedorf, Switzerland). The negative genetic assay was used for the initial screening of compounds. The screening was done in 96-well plates, of which the first and last columns were used for control with 1% DMSO. The bacterial cells without IPTG were taken as a negative control, while bacteria with 100 µM IPTG were taken as the positive control.

Selected compounds from the first screening were examined with the positive genetic assay in 10 µM IPTG and 5 mM KCl conditions. Assay for proton flow was performed as earlier mentioned [30,38,43,44,45] with an exception, whereby the cells were treated with 100 µM of compounds before applying citric acid (300 mM, 0.9% NaCl). Uninduced cells were used as the negative control. Reading of buffer was deducted to cancel background noise. 1% DMSO was used in all control reactions for the compounds.

## 3. Results

### 3.1. Sequence Alignment of Flavivirus Proteins

The sequence alignment of membrane glycoprotein ZikV-M (YP-009227197) with Dengue of membrane glycoprotein (NP-722459.2) and West Nile matrix protein M (NP-776013.1) exhibited a conserved region of amino acids (Appendix A). We have chosen several of the conserved amino acids of the transmembrane region (Appendix A) for further mutagenesis studies, as expounded below.

### 3.2. Protein Expression and Membrane Integration

In order to confirm protein expression of the cloned constructs, a Western blot was performed (Appendix A). The Western blot depicted a band around 50 kDa for ZikV-M, WNV MgM, and DenV MgM composed of the 42 kDa corresponding to the maltose-binding protein and 8 kDa arising from the viral protein. A higher band also present in the other flavivirus membrane proteins may indicate a partial oligomeric species.

Targeting the protein into the internal bacterial membrane and proper integration was confirmed by transforming the plasmid into NT326 cells [46,47,48]. Such cells lack a native MBP and consequently cannot grow on M9 media with maltose as the only carbon source. However, a plasmid-based MBP directs protein expression to the inter-membrane space and allows the bacteria to grow on M9 maltose [46,47,48]. As shown in Appendix A, NT326 cells with MBP-ZikV-M construct grew on M9 media supplemented with 1% maltose after 72 h at 37 ${}^{\circ }$C. Taken together, the said construct results in protein expression and integration in the inner bacterial membrane.

### 3.3. Characterization of Ion Channel Activity

#### 3.3.1. Negative Genetic Assay

The negative assay is based on the fact that elevated protein expression levels result in growth retardation due to membrane permeabilization by an active viroporin. This assay is well established for the characterization of viroporins in earlier studies [30,39,41,42,43,44,45,55]. Indeed, ZikV-M, when expressed at increasing levels, showed commensurate bacterial growth inhibition, as shown in Figure 1.

#### 3.3.2. Positive Genetic Assay

Other spurious factors can cause growth retardation in bacteria by heterologous protein expression. Therefore, to further validate the channel activity of the protein, a second assay was performed to complement the first assay. The assay is called a positive genetic assay and is well characterized for identifying viroporins [30,41,42,43,44,45]. In general, this assay is specific for the K^+^ conductivity of channels. LB650 cells are K^+^-uptake deficient bacteria that exhibit retarded growth in low potassium media (i.e., LB) [36].

The ZikV-M construct when transformed into K^+^-uptake deficient bacteria and upon induction with IPTG, resulted in improved bacterial growth, most likely due to the potassium channel conductivity of ZikV-M. As seen in Figure 2, the bacterial growth was improved up to 20 µM IPTG. Beyond that, the growth started to slow down due to excess membrane permeabilization, as seen in other viroporins [30,41,42,43,44,45].

#### 3.3.3. Proton Conductivity Test

A fluorescent-based conductivity assay was used to examine if ZikV-M can facilitate proton flow. LR1 bacteria harbor a pH-sensitive green fluorescent protein [37] in their genome [38]. Subsequently, injection of an acidic solution to the media will result in a detectable fluorescence change if the bacteria express a protein capable of proton transport.

As seen in Figure 3, uninduced bacteria do not show appreciable proton flow. However, when induced with 100 µM IPTG, a significant increase in proton flow and reduced pH inside the cell is observed. Hence, elevated ZikV-M protein expression results in the activation of proton channel activity.

### 3.4. Mutagenesis

The sequence alignment of ZikV-M with known viroporins from Dengue virus 1 and West Nile virus was able to outline the conserved region of the transmembrane region (Appendix A). The modeled structure (Appendix A) showed the transmembrane domain of the ZikV-M protein with three -helices. It has been predicted that helix 2 and 3 constitute the transmembrane region of the protein, while helix 1 is responsible for the interaction with the E protein [35].

We have focused mainly on conserved amino acids from the aforementioned flaviviruses for mutagenesis. All mutations were done by replacing the analyzed residue with an alanine. The selected positions were P40, G41, G54, Q59, K60, L69, and P72. Amino acids P40 and G41 are located at the beginning of helix 2, while G54 lies in the connecting loop between helix 2 and 3. Residues Q59 and K60 are positioned at the beginning of helix 3. We have introduced a stop codon at position 61 to understand whether helix 3 has an important role in viroporin activity. Residues L69 lies in helix 3, whereas P72 is positioned outside the transmembrane region of the protein (Appendix A). Subsequently, the channel activities of each of the mutants and wild-type proteins were examined in each of the three bacteria-based assays as expounded below.

#### 3.4.1. Negative Genetic Assay

In order to establish the role of the conserved amino acids, we performed the negative assay with the designed mutants. All mutants were transformed into DH10B cells and treated with increasing IPTG concentrations from 25 to 200 µM. As shown above, in Figure 1, elevated expression of wild-type ZikV-M was deleterious to bacterial growth. Correspondingly, the impact of the different mutations on bacterial growth is depicted in Figure 4A (Appendix A). Two mutations abrogated the ability of the protein to retard growth: G54A (in the connecting loop between helix 2 and 3) and Q59A (helix 3). Mutant P40A (helix 2) and V61Stop also reduced, albeit to a low extent, the inhibition of bacterial growth relative to the wild-type protein.

In order to investigate the role of some of these mutations in corresponding flavivirus proteins, we performed the negative assay using G54A and P61Stop of WNV and DenV MgM viroporins. In both flavivirus proteins, the G54A mutation exhibited only a mild impact on the ability of the WNV and DenV viroporins to retard growth. This result is in stark contrast to the impact of the mutation in ZikV-M, which neutralized the protein entirely.

A different result was obtained after inserting a stop codon at residue 61. While ZikV-M and DenV MgM exhibited only a mild impact, in WNV, it inhibited the protein almost entirely (Figure 4B,C, Appendix A).

#### 3.4.2. Positive Genetic Assay

We performed the second assay to corroborate the results from the negative genetic assay. ZikV-M mutants were transformed into K^+^-uptake deficient bacteria and the corresponding bacteria were grown at low potassium media (5 mM) by inducing the channel (10 µM IPTG). As expected, the wild-type protein enables the bacteria to thrive upon induction. However, mutants P40A, G54A, Q59A, and V61Stop reduce the ability of bacteria to grow in the aforementioned conditions. These results again confirmed the importance of these amino acids to the channel activity of studied viroporin (Figure 5, Appendix A).

We subsequently performed the assay for WNV and DenV MgM to compare their behavior with ZikV-M. In this instance, a stop codon at residue 61 inhibited the activity of both proteins appreciably. However, the G54A mutation only mildly impacted WNV MgM and exhibited no activity on the corresponding protein from DenV (Appendix A).

#### 3.4.3. Proton Conductivity Test

To determine the effect of the mutations on proton conductivity, we utilized the third fluorescent-based assay (Figure 3A). Wild type ZikV-M and all mutants were grown up to three hours and were then induced by 100 µM IPTG for two hours. Uninduced ZikV-M was taken as control. All mutations exhibited less proton flow than the wild-type protein. Mutants G54A, Q59A, and V61-stop exhibited less channel activity comparable to uninduced wild type and were much lower than IPTG induced wild type. The result once more pointed at the crucial role of G54, Q59, and helix 3 in channel activity.

Like ZikV-M, the G54A and 61 stop mutations were examined in WNV MgM and DenV MgM. In WNV, both mutations reduced proton flow appreciably. In DenV, MgM mutants G54A showed reduced flow while 61-stop did not show much reduction in the activity (Figure 3B, Appendix A).

### 3.5. Drug Screening

After establishing the channel activity of the ZikV-M protein, we sought to identify blockers against the channel. For this purpose, two libraries of compounds were screened. The first contained 372 compounds in the area of “Membrane Transporter/Ion Channel”, while the second was a repurposing library with 2839 compounds. The method used for screening in this study was already established in earlier studies for channels in SARS CoV-2 [43,44,45]. In brief, each compound was first tested in the negative assay, where the viral channel is detrimental to bacterial growth due to membrane permeabilization. Therefore, in this assay, positive hits are identified based on their ability to revive bacterial growth.

The compound concentration used for screening was 100 µM. Our rationale for a high screening concentration stemmed from our intent to cast large a net as possible, which is afforded by the bacterial system. Any compound identified in the screen might serve as a starting point for further exploration, yielding more efficacious hits.

As seen in Figure 6A, ten compounds were able to increase bacterial growth in the negative assay to varying extents: Floxuridine, Nafamostat, Kasugamycin (hydrochloride hydrate), 5-Azacytidine, AZD-5423, Streptomycin (sulfate), Zosuquidar, Dequalinium chloride, Capreomycin (sulfate), and Pentamidine (isethionate).

The above ten compounds were subsequently examined in the positive genetic assay in K^+^-uptake deficient bacteria. In this assay, ZikV-M enhances the growth of the bacteria, and therefore, blockers are expected to reduce growth. Gratifyingly, all of the compounds that scored positively in the first assay were able to decrease bacterial growth, with the exception of Nafromastat and Zosuquidar (Figure 6B).

Finally, the proton flow assay examined the ten compounds that tested positively in the negative assay. In this instance, the internal pH of bacteria changes upon injection of acid to the media if the bacteria express a channel capable of proton transport. Therefore, successful blockers will inhibit the aforementioned acidification, which is detected by the fluorescence change [38] of a pH-sensitive GFP [37]. Results shown in Figure 6C demonstrate that all compounds were indeed able to decrease H^+^ conductivity to varying extents except for Dequalinium chloride.

## 4. Discussion

The prevalence of pathogenic viruses and limited treatment options for the diseases they cause has led to in-depth research towards understanding the physiology and pathogenic mechanism of these etiological agents. The genome of several pathogenic viruses encodes for hydrophobic integral membrane proteins spanning 50–120 amino acids long, termed as viroporins. These proteins have become the subject of much interest owing to their role in the viral infectivity cycle.

Viroporins are encoded by a range of viruses of clinical interest [27,56], such as 2B of poliovirus [57], 6k of alphaviruses [58], 2k of dengue virus [30], E and 3a from SARS CoV-2 [43,44], hepatitis C virus p7 [59,60], and many others. However, the M2 proton channel from the Influenza A virus remains the best-characterized viroporin with provenance as an antiviral drug target. Specifically, the anti-flu drugs amantadine or rimantadine exert their anti-flu activity [61] by inhibiting M2 [62] through the blockage is its channel activity [28].

Flaviviruses are vector-borne enveloped RNA viruses assembled using three viral structural proteins (C, prM, and E), a host lipid envelope, and the viral genomic RNA. The most prevalent flaviviruses which can emerge as significant public health threats within a short period include DenV, WNV, and ZikV. The small integral membrane (M) from these pathogenic viruses have been studied for its ion channel activity and subsequently characterized as viroporin [30,56,63,64].

In the present study, we have investigated the viroporin activity of the M protein from ZikV. In addition, we examined the importance of several of the conserved amino acids of its transmembrane domain and compared them to other flavivirus viroporins. Finally, we employed high-throughput screening for drug identification of ZikV-M blockers.

The gene corresponding to ZikV-M was cloned into the pMAL-p2X vector generating a chimera with MBP, suitable for targeting the protein to the bacterial inner membrane. This construct has been used successfully with numerous other viroporins [30,39,42,43,44,55]. Western blotting established protein expression (Appendix A), and membrane incorporation could be confirmed with the maltose complementation assay (Appendix A).

In order to characterize the channel activity of the protein, we made use of three well-established and independent bacterial-based assays [30,39,42,43,44,45,55]. The first two assays are reciprocal to each other: In the negative genetic assay, elevated protein expression is responsible for retarded bacterial growth due to excessive membrane permeabilization (Figure 1). However, in the positive genetic assay, protein expression enables K^+^-uptake deficient bacteria to survive in low K^+^ media (Figure 2). Hence, these two assays reduce the probability of spurious factors influencing the results. The third fluorescence-based assay was conducted to confirm H^+^ conductivity by measuring the fluorescence change due to acidification of the external media employing a pH-sensitive GFP [37] chromosomally expressed in the bacteria [38] (Figure 3).

The results from these three assays substantiate the ion channel activity of the ZikV-M. Our experimental flow showed similar results to that of M2 from IVA [39,41,55], Vpu from HIV [42], E [44] and 3a [43] from SARS CoV-2, gp151 and gp170 from variola virus [30], MgM and 2k from DenV [44], MgM from WNV [44], and 6k from Eastern equine encephalitis [44]. Hence, our results point to ZikV-M as a viroporin that can conduct both K^+^ and H^+^

The sequence alignment of ZikV-M shows overall 45.33% and 34.67% identity with WNV MgM and Dengue 1 MgM, while the transmembrane region of ZikV-M had 61.11% and 33.33% identity, respectively (Appendix A). Our mutational studies using all three bacterial-based assays showed that introducing a stop codon at position 61 reduced the ion channel activity in all three flaviviruses M proteins studied (Figure 3 and Figure 4). These results suggest that helix 3 may have an important role in forming the transmembrane domain of these viroporins.

Moreover, two ZikV-M mutants, G54A and Q59A, were able to affect the channel activity of ZikV-M appreciably. Interestingly, the G54A mutation was less perturbing to channel activity of WNV MgM and DenV MgM (Figure 3 and Figure 4). Our results correlate with previous studies [34,35,65], where the authors have mentioned that helixes 2 and 3 from the transmembrane domain in the viral membrane. The N–terminus of the protein interacts with the E protein, which rearranges into a trimeric fusion complex and releases matured M protein which eventually forms oligomers and functions as an ion channel.

Based on previous directions, viroporins are emerging as potential targets for clinical intervention [27,56,66]. In this direction, identifying new viroporins and developing high-throughput screening systems may facilitate the discovery of new inhibitors of viral infection that can curb future viral diseases. However, amantadine and rimantadine are the only proven antiviral drugs inhibiting ion channel activity of viroporins [61,67,68].

Therefore, after establishing that ZikV-M as viroporin in our experimental assays, we focused on identifying potential drugs against it by screening drug libraries, including ion-channel inhibitors and approved drugs for human usage. Our hope was to fast-track the drug discovery process by drug repurposing using three bacterial-based assays [69,70,71,72].

Our combined approach yielded ten compounds that enhanced the growth of bacteria that are otherwise suffering from viroporin membrane permeabilization (Figure 6). Of these compounds, eight inhibited the growth of K^+^-uptake deficient bacteria dependent on the viroporin’s activity. Furthermore, all but one of the ten compounds could also reduce viroporin-driven H^+^ flow.

In conclusion, our findings provide a list of blockers of ZikV M protein that should be examined in an in vitro and in vivo setting to establish their potential as antiviral agents. Such agents may also prove helpful in exploring the complex function of the channel in the infectivity cycle of the protein.

## Figures and Tables

**Figure 1 biomedicines-10-00641-f001:**
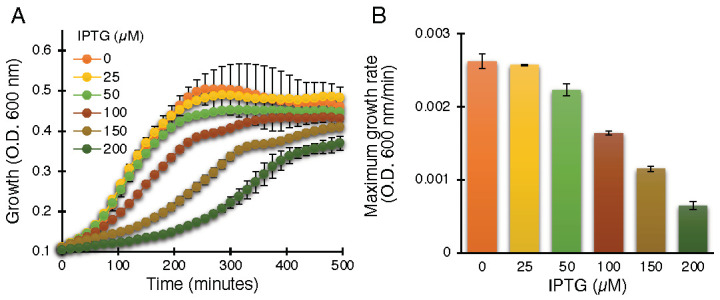
Standardization of ZikV-M as a viroporin in the negative genetic assay. Growth curves (**A**) and maximal growth rates (**B**) of DH10B bacteria expressing ZikV-M, grown in LB media with an increasing dose of IPTG, as noted.

**Figure 2 biomedicines-10-00641-f002:**
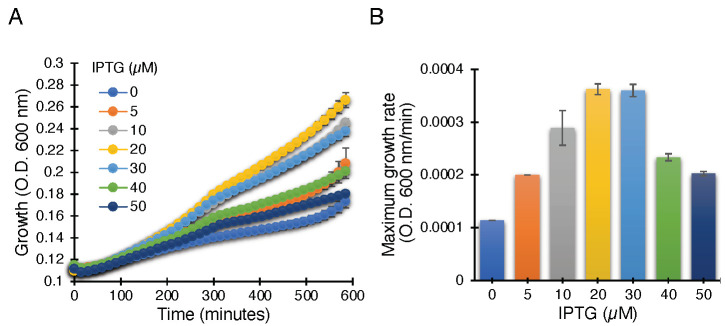
Positive genetic assay utilizing K^+^-uptake deficient bacteria, which are unable to grow on LB media without potassium supplement. Growth curves (**A**) and maximal growth rates (**B**) of K^+^-uptake deficient bacteria harboring the ZikV-M plasmid grown with different concentrations of IPTG, as noted.

**Figure 3 biomedicines-10-00641-f003:**
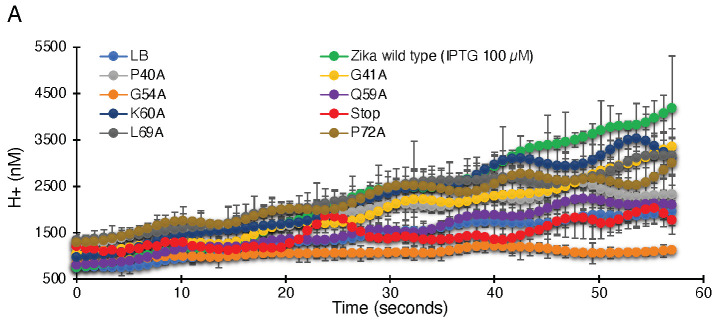
Proton conductivity of flavivirus viroporins and mutants thereof. Wild-type ZikV-M, WNV MgM, DenV MgM, and mutants were transformed into bacteria that harbor a pH-sensitive green fluorescence protein [37,38]. Proton flow for uninduced wild type, induced wild type, and mutants (100 µM IPTG) were recorded after two hours of induction. Panel (**A**) depicts the H^+^ concentration change as a function of time, while the slopes of the individual curves are listed in panel (**B**).

**Figure 4 biomedicines-10-00641-f004:**
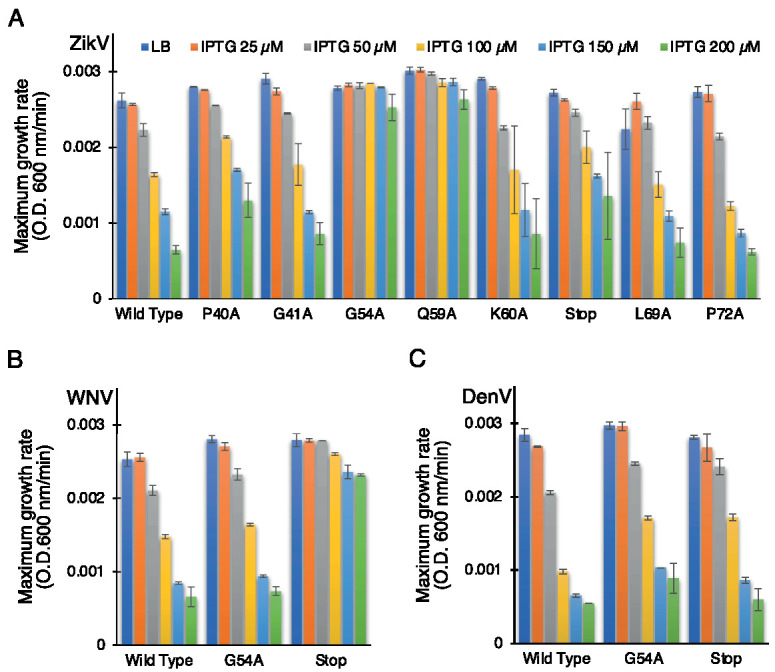
Effect of mutants of flavivirus membrane proteins in the negative genetic assay. Maximal growth rates of bacteria that express mutants of ZikV-M (**A**), WNV MgM (**B**), and DenV MgM (**C**) as a function of increasing IPTG, as noted.

**Figure 5 biomedicines-10-00641-f005:**
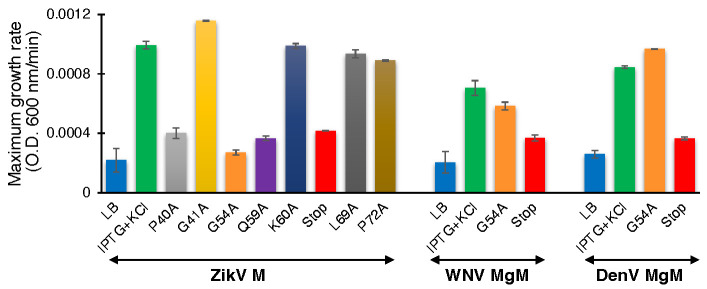
Effect of mutants of flavivirus membrane proteins in the positive genetic assay. Maximal growth rates of K^+^-uptake deficient bacteria that express mutants of ZikV-M, WNV MgM, and DenV MgM grown with 10 µM IPTG.

**Figure 6 biomedicines-10-00641-f006:**
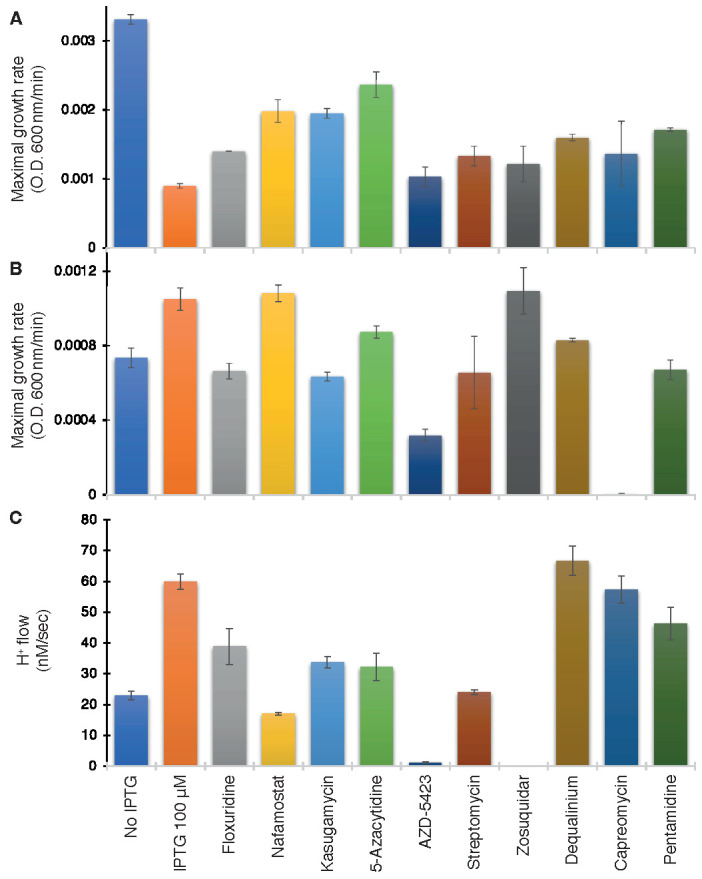
Drug screening on wild-type ZikV-M. Maximal growth rate as a function of the different drugs (100 µM) in the negative genetic assay (**A**) or positive (**B**) assays. (**C**) Effect of drugs on proton flow examined on bacteria that express a pH-sensitive GFP [38]. Note, that effective compounds are expected to increase the growth rate in the negative assay (**A**), decrease growth in the positive assay (**B**), and decrease H^+^ flow in the pH-assay (**C**).

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
