# Peer review of "Zika M—A Potential Viroporin: Mutational Study and Drug Repurposing"

_biomedicines, 2022, doi:10.3390/biomedicines10030641_

Round 1

Reviewer 1 Report

This report supports countermeasures against the flavivirus genus including Zika virus, which involves a major health concern.

The authors studied M protein using an authentic E. coli system. M protein production and functional analysis were performed by examining functional complementarity. Ten compounds were identified as ZikV-M blockers. The results of the study can be the basis for building high-throughput screening systems.

This paper has a valid logical structure, and the experimental results are shown with statistics.

Author Response

We thank the reviewer for the comments. We proofread the manuscript rigorously to ensure the omission of English errors

Reviewer 2 Report

The manuscript titled "Zika M - A potential viroprotein: Mutational study and drug repurposing" by Arkin and co-workers is well written and should be considered for publication after the authors address the following minor issues:

  1. What were the inclusion-exclusion criteria for the drugs used in repurposing studies?
  2. The authors should specify molecular weight range for the compounds tested during drug screening. What percent of these compounds were approved as drugs?
  3. Since the repurposing studies were done at high concentration (100 micromolar), the authors should comment on the potential toxicity of the compounds at such high concentration.

Author Response

1. We did not construct the repurposing drug library. Rather, we purchased the largest library that we could find from a reputable commercial vendor and used it as is. According to the vendor (MedChem Express) "MCE Drug Repurposing Compound Library contains 3775 approved drugs and passing phase I clinical drugs, which have been completed extensive preclinical and clinical studies and have well-characterized bioactivities, safety and bioavailability properties". We now mention this in detail in the manuscript. Note that the number of chemicals in the library has increased since we purchased it.

2. As stated above, we have used the library as is. The average molecular weight is 430 Da with a standard deviation of 277 Da. These values are now specified in the article. The list of compounds and their description is available on the vendor's website and is once more mentioned in the manuscript.

3. Our rationale for a high screening concentration stemmed from our intent to cast a large net as possible, which is afforded by the bacterial system. Any compound that is identified in the screen might serve as a starting point for further exploration yielding more efficacious hits. We include this discussion in the manuscript.